# Hydrodynamic Responses and Machine Learning-Based Shape Classification of Harbor Seal Whiskers in the Wake of Bluff Bodies

**DOI:** 10.3390/biomimetics10080534

**Published:** 2025-08-14

**Authors:** Xianghe Li, Zhimeng Zhang, Hanghao Zhao, Yaling Qin, Muyuan Du, Taolin Huang, Chunning Ji

**Affiliations:** 1State Key Laboratory of Hydraulic Engineering Intelligent Construction and Operation, Tianjin University, Tianjin 300354, China; lxh_2001@tju.edu.cn (X.L.); hh_zhao99@tju.edu.cn (H.Z.); dmy1397726976@tju.edu.cn (M.D.); taolinhuang@tju.edu.cn (T.H.); cnji@tju.edu.cn (C.J.); 2MOE Key Laboratory of Marine Intelligent Equipment and System, Shanghai Jiao Tong University, Shanghai 200240, China; sjtu_qyl@sjtu.edu.cn

**Keywords:** fluid–structure interaction, seal whisker, machine learning, shape classification, generalization ability

## Abstract

Harbor seals, equipped with their uniquely structured whiskers, demonstrate remarkable proficiency in tracking the trajectories of prey within dark and turbid marine environments. This study experimentally investigates the wake-induced vibrations of an elastically supported whisker model placed in the wakes of circular, square, and equilateral triangular cylinders of varying dimensions. Thereafter, a machine learning model is trained to identify and classify these intrinsic responses. The findings reveal a positive correlation between the amplitude of vibration and the total circulation shed by the bluff bodies. In the wake flow fields of triangular and circular cylinders, the mean drag is quite similar. Meanwhile, the whisker’s vibration amplitude and drag fluctuation show that the triangular cylinder is comparable to the square cylinder, and both are higher than the circular cylinder. To classify the wake-generating body shapes based on the hydrodynamic characteristics, hydrodynamic features encompassing vibration amplitudes, fluid forces, and frequency-related information were extracted to train an LSTM-based model, and it was found that the mean drag significantly enhances the model’s flow velocity generalization performance.

## 1. Introduction

Harbor seals and other pinnipeds have evolved remarkable capabilities to hunt effectively in dark, turbid, and complex marine environments. This remarkable capability is predominantly ascribed to the exquisite sensitivity of their whiskers to hydrodynamic signals [1]. Even when deprived of hearing and sight, harbor seals can utilize their whiskers to discern subtle differences, such as distinguishing between disks of varying sizes [2,3], identifying a single vortex ring [4], and detecting the breathing currents emitted by fish through their gills [5,6,7]. Miersch et al. [8] measured the vibration of an isolated single harbor seal whisker in the wake of a cylinder, confirming that the harbor seal whisker exhibits a high signal-to-noise ratio (SNR) in wake identification, highlighting its potential as a signal recognition sensor. The exceptional vortex-sensing capabilities of harbor seal whiskers have attracted considerable attention from researchers, who are striving to develop stealthy, energy-efficient, and environmentally friendly biomimetic underwater sensors inspired by these natural adaptations.

To elucidate the underlying mechanisms behind the remarkable capabilities of harbor seal whiskers, extensive research on the whiskers’ structural features and hydrodynamics has been conducted in uniform flow. Hanke et al. [9] employed micro-photography techniques to meticulously measure the shape of harbor seal whiskers, revealing their specialized undulated surface structure. They identified seven characteristic parameters to represent the structural features of the whisker surface. These parameters include the four semi-axes of the two controlling ellipses (*a*, *b*, *k*, *l*), the distance between the two ellipses (*M*), and the attack angles of the two ellipses (*α*, *β*). Then, Hanke et al. [9] utilized numerical simulations to visualize the wake structure behind the whiskers at Reynolds numbers of 300 and 500. They discovered that the unique wavy shape of harbor seal whiskers significantly suppresses vortex-induced vibrations (VIVs), leading to reduced background noise and thereby enhancing their ability to identify wakes.

In underwater signal sensing, vortex shedding from bluff bodies is a common phenomenon observed in both natural environments and engineering applications. To fully exploit the signal identification capabilities of harbor seal whiskers, numerous researchers have employed the whisker model proposed by Hanke et al. [9] (or its numerically modified versions [10,11]) to investigate the hydrodynamic characteristics of whiskers in the wake of bluff bodies.

In the wake of the cylinder, Beem et al. [12] conducted experimental studies on the vibration response of an elastically supported whisker model. They found that the non-dimensional amplitude of the whisker model increased significantly, and the dominant frequency of vibration aligned with the vortex shedding frequency of the upstream cylinder. Zheng et al. [13] compared the responses of the whisker model array with and without a cylinder placed in front. Their results indicated that the whisker sensor possesses an extremely high signal-to-noise ratio and can efficiently detect long-range wakes. Song et al. [7,14,15] employed numerical simulations and experiments to study the hydrodynamic characteristics of elastically supported single-degree-of-freedom vibrating harbor seal whiskers in the cylindrical wake. They observed that within the wake flow field, the vibration response of the whisker structure was significantly enhanced, exhibiting stability and periodicity. The amplitude was positively correlated with the size and intensity of the wake vortices, and the vibration frequency of the whisker model was locked to the vortex shedding frequency of the wake flow field.

For other bluff body shapes, Shan et al. [16,17] performed numerical simulations of vortex-induced vibrations of whiskers under the wakes of cylindrical, square, and diamond-shaped columns at a Reynolds number of Re = 300. It revealed that under different wakes, the main frequency of the whisker’s transverse vibration consistently matched the vortex shedding frequency of the upstream object, while there were significant differences in the flow patterns between the columns and related vibration statistical parameters. Using machine learning methods, Mao et al. [18] conducted experiments to measure the forces on the harbor seal whisker model under nine different upstream bluff body conditions and then adopted a convolutional neural network (CNN) for training and testing the force signals. Their results demonstrated that, in most cases, the trained model could successfully identify objects in water.

Although previous studies have explored the wake-induced vibrations of harbor seal whisker models and applied machine learning to classify upstream bluff body shapes, they have primarily focused on identifying maximum or minimum response characteristics under specific flow conditions for each shape. However, the dynamic interactions and underlying mechanisms governing wake-induced whisker responses vary with flow velocity and upstream flow conditions, even for identical shapes. Consequently, shape identification requires a comprehensive analysis of vibration and hydrodynamic characteristics to ensure robust generalization and accuracy, avoiding reliance on localized features. Moreover, directly using displacement or fluid force signals for machine learning training retains all information but easily leads to overfitting due to irrelevant data. This also makes it hard to identify significant features for recognizing bluff body shapes, limiting our understanding of how harbor seals distinguish different shapes. Considering that practical deployment of seal-inspired whisker sensors would involve complex and unsteady flow environments, the generalization capability of classification models becomes critically important. Despite this need, limited attention has been given to evaluating or improving the generalization performance of such models. Therefore, the present study systematically investigated the hydrodynamic responses of an elastically supported, single-degree-of-freedom whisker model in the wakes of different bluff bodies. Subsequently, an LSTM-based classification model is employed to identify the shape of the upstream bluff body using the extracted features, rather than raw time-series data. This approach focuses on assessing the model’s generalization performance across varying flow velocities.

## 2. Materials and Methods

### 2.1. Experimental Setup

To explore the wake-induced vibration mechanisms and create a dataset for machine learning-based shape identification, upstream objects were modeled as bluff bodies of different shapes and sizes (circular, square, and equilateral triangular cylinders) to generate distinct wake patterns. The experiment was conducted in a low-turbulence recirculating water flume located in the Fluid Mechanics Laboratory at Tianjin University. The vibration responses of the seal whisker model were measured using the Low-Damping Vibration Measuring System (LODVMS), as shown in Figure 1. The test section of the water flume had a working area of 400 mm (height) × 306 mm (width) × 2370 mm (length). The water depth was maintained at 330 mm. The whisker model is rigidly fixed at its root to the slider of an air-bearing track. The slider is supported by springs at both ends, with the track restricting its motion exclusively to the cross-flow direction, resulting in single-degree-of-freedom vibration. Notably, the whisker model utilized in this experiment is characterized by complete rigidity, exhibiting no elastic deformation. To prevent complex tip vortex shedding, an end plate measuring 2 mm (thickness) × 200 mm (width) × 300 mm (length) was positioned horizontally beneath the whisker model [19].

The Reynolds number (Re), reduced velocity (*U_r_*), mass ratio (*m**), and structural damping ratio (*ζ*) are defined as Re = *U_∞_D*/*ν*, *U_r_* = *U_∞_*/*f_n_D*, *m** = *m_s_*/*m_f_*, and *ζ* = *c*/2√(*km_s_*), respectively, where *U_∞_* is the inflow velocity, *D* is the narrow side mean diameter of the whisker model, *ν* is the kinematic viscosity of water, *f_n_* is the structural natural frequency in still water, *m_s_* is the vibration system mass, *m_f_* is the displaced fluid mass, *c* is the damping coefficient, and *k* is the stiffness. The key values for the present experiments are listed in Table 1.

The model proposed by Hanke et al. [9] was enlarged by a factor of 1:30 in this experiment. The physical model is shown in Figure 2. The model parameters are *M* = 27.3 mm, *a* = 17.85 mm, *b* = 7.2 mm, *k* = 14.25 mm, *l* = 8.7 mm, *α* = 15.27°, and *β* = 17.6°. The narrow side mean diameter of the whisker model is *D* = 2 × (*b* + *l*)/2 = 15.9 mm.

The upstream bluff bodies were chosen in three shapes: circular cylinder, equilateral triangular cylinder, and square cylinder (as shown in Figure 3). For each shape, three characteristic dimensions were selected: 16 mm, 32 mm, and 48 mm, respectively (≈1*D*, 2*D*, 3*D*). For the circular cylinder, the characteristic dimension is the diameter. For the square cylinder and equilateral triangular cylinder, the characteristic dimension is the side length of the cross-sectional shape. The specific arrangement angles of the prisms are shown in Figure 4. The distance between the upstream bluff bodies and the whisker model was set at 10D (=159 mm) to ensure that the whisker model could effectively sense the strong vortex signals generated by the upstream bluff bodies.

### 2.2. Verification of the Experimental Setup

To ensure the reliability of the experimental results, the experimental setup was validated by comparing the results with Khalak and Williamson [20] for the vortex-induced vibration (VIV) of an isolated circular cylinder with one degree-of-freedom. The lift coefficients are calculated and nondimensionalized as(1)$$C_{L}=\frac{2F_{L}}{\rho U_{\infty }^{2}DL}$$
where *F_L_* is the total lift force of the model, *ρ* is the fluid density, and *L* is the submerged length of the whisker model.

As shown in Figure 5, the validation experiment (*m** = 3.39, *ζ* = 0.0019) effectively replicated the VIV branches documented by Khalak and Williamson [20] (*m** = 3.3, *ζ* = 0.0026). The vibration amplitudes, vibration frequencies, and root mean square (rms) lift coefficients in the validation experiment were in close agreement with those reported by Khalak and Williamson [20] across the initial, upper, lower, and desynchronization branches. This consistency with prior research confirms the accuracy of the current experimental setup.

### 2.3. LSTM Structure and Settings

Long Short-Term Memory (LSTM) [21] is a special type of Recurrent Neural Network (RNN) designed to process and predict time series data. LSTM, through its unique structure and internal mechanisms such as the cell state, hidden state, input gate, output gate, and forget gate, effectively addresses the gradient explosion and long-term dependency problems that exist in traditional neural networks [22].

Figure 6 and Table 2 provide the structure and parameters of the LSTM-based classification model used in this study. The parameters were determined through the Runge–Kutta optimization algorithm [23] to achieve optimal performance. It consists of a single LSTM layer followed by a ReLU activation layer and a fully connected layer with a softmax output for classification. Specifically, the LSTM layer has 100 hidden units and operates in the ‘last’ mode, outputting only the hidden state from the final time step of the input sequence. This design enables the model to effectively capture long-term dependencies within the sequence of data. The ReLU activation layer introduces non-linearity, enhancing the model’s ability to learn complex patterns. The fully connected layer maps the LSTM output to the classification space, with the softmax function converting the output into a probability distribution over the target classes. The input to the network is extracted hydrodynamic features, rather than raw time-series signals. The input size of the network is dynamically determined by the number of selected feature numbers. The output of the model is a categorical prediction representing the shape of the upstream bluff body. The model directly outputs the predicted class label, and the category with the highest probability is selected as the predicted classification result.

## 3. Results and Discussion

The vibration responses, as well as lift and drag forces and their corresponding frequency characteristics exhibited by the harbor seal whisker model, demonstrate a strong dependence on its interaction with the wake flow generated by upstream bluff bodies. In this section, the experimental data are first analyzed to examine the vibration responses and fluid forces experienced by the whisker model in the wakes of various bluff bodies. Subsequently, hydrodynamic features are extracted for training an LSTM-based classification model, and the model’s flow velocity generalization performance is investigated.

### 3.1. Hydrodynamic Responses

#### 3.1.1. Vibration Responses

As shown in Figure 7, the trends of the non-dimensional vibration amplitudes *A**_10_ (*A**_10_ = *A*_10_/*D*, where *A*_10_ denotes the top 10% of the maximum amplitudes) versus the reduced velocity *U_r_* (*U_r_ = U_∞/_f_n_D*) are presented for the whisker model positioned in the wake of different upstream bluff bodies. As depicted in Figure 7a–d, the wakes of these bluff bodies excite pronounced vibrations in the whiskers, with *A**_10_ reaching 3 to 30 times the values observed in uniform flow conditions. Furthermore, *A**_10_ consistently follows a trend of initially increasing and then decreasing with *U_r_*. This behavior aligns with the findings of Beem et al. [12] regarding whisker vibrations in the wake of a circular cylinder.

Furthermore, Figure 7 highlights the substantial impact of the characteristic dimensions and shape of the bluff body on *A**_10_, with a summary of these trends presented in Figure 8. A clear positive correlation exists between characteristic dimensions and the peak values of *A**_10_. This correlation arises because larger bluff bodies generate stronger and more persistent vortex structures in the wake, as evidenced by higher vortex strength and slower decay along the flow direction. Moreover, the shape of the bluff body modulates the wake’s vortex shedding frequency and pattern, intensifying the disturbance on the whisker and amplifying its vibrational response.

The characteristic peak *A**_10_ values exhibit striking shape dependence, with square and triangular cylinder wakes producing vibrations 70% stronger than circular cylinder cases (Figure 8). This pronounced contrast emerges from fundamental differences in flow separation patterns: while the sharp corners of square/triangular cylinders fix separation at *θ* = 45°, circular cylinders exhibit delayed separation (*θ* ≈ 80°) [24]. Such geometric control of separation directly modulates the resultant vortex dynamics—a dimension rarely explored in previous whisker vibration studies that predominantly considered circular cylinders. According to the theory of Fage and Johansen [25], a smaller flow separation angle (*θ*) results in greater streamline deflection and causes the separated shear layers to move further away from the bluff body. This leads to a larger separated shear layer velocity. Roshko [26] proposed the following equation to estimate the total circulation shedding from a cylinder:(2)$$\frac{\Gamma_{0}}{U_{\infty }d}=\frac{U_{s}^{2}T}{2U_{\infty }d}=\frac{1}{2St}{\left(\frac{U_{s}}{U_{\infty }}\right)}^{2}$$
where Γ_0_ is the total circulation, *T* is the vortex shedding period, *St* is the cylinder’s Strouhal number, and *U_s_* is the separated shear layer velocity. It can be observed that Γ_0_ is directly proportional to *U_s_*. Therefore, the wakes of square and triangular cylinders exhibit greater total circulation compared to the wake of a circular cylinder, resulting in a more pronounced response from the whiskers. For a circular cylinder, the maximum vorticity of the shedding vortex is attenuated compared to a square cylinder, and the circulation associated with the vortices decreases progressively by up to 50% [27].

To comprehend how upstream vortex shedding affects the vibrational response of the whisker model, this study investigates the correlation between the dominant vibration frequency and the vibration amplitude. Figure 9 highlights this correlation by showing the relationship between the non-dimensional dominant vibration frequency *f*_dominant-Y_* (*f ** = *f*/*f_n_*) and *A**_10_ using a *D* = 48 mm triangular cylinder as an example. The dominant vibration frequency of the whisker model increases linearly with *U_r_*, fitting to *St* = 0.165. This finding demonstrates that the whisker model can effectively detect the vortex shedding frequency from the upstream triangular cylinder, which is a typical “wake-induced” phenomenon [28], as also found by Shan et al. [17] Additionally, the natural frequency of the vibration system (*f** = 1) is also evident in the amplitude spectrum. When the dominant vibration frequency of the model approaches the natural frequency, *A**_10_ rapidly increases and reaches its peak. This behavior can be attributed to structural resonance [29]. The whisker model consistently exhibits these characteristics across various wake conditions. This consistency explains why *U_r_*, at which the whisker model reaches its peak *A**_10_, varies in different upstream bluff body wakes, as shown in Figure 7. The diameter and shape of the bluff body significantly influence the vortex shedding frequency. A smaller characteristic dimension (or a higher *St*) of the upstream bluff body results in a higher vortex shedding frequency at the same *U_r_*. Consequently, the vortex shedding frequency reaches the structural natural frequency at a lower *U_r_*, thereby maximizing *A**_10_.

#### 3.1.2. Fluid Forces

The variation in the root-mean-square (RMS) lift forces (*F_L,rms_*) acting on the whisker model is depicted in Figure 10 across different experimental conditions. Overall, *F_L,rms_* increases with the increasing *U_r_*, reflecting the enhanced interaction between the whisker and the stronger, more energetic vortex structures in the wake at higher flow velocities. However, within specific regions (marked by the red dashed box), *F_L,rms_* exhibits a local peak followed by a valley, forming an “N-shaped” pattern. This phenomenon is evident in most cases, except in the 32 mm circular cylinder wake, where it is not pronounced. Additionally, it is notable that the inflection point of *F_L,rms_* occurs at *U_r_*, which closely corresponds to the positions where *A**_10_ reaches its peak.

To explain this phenomenon, Figure 11 shows the non-dimensional dominant frequency of vibration (*f*_dominant-Y_*) and lift force(*f*_dominant-FL_*) within the inflection range of *F_L,rms_*, taking the 16 mm triangular cylinder as an example. It can be seen that *f*_dominant-Y_* and *f*_dominant-FL_* coincide at most *U_r_*. However, separation occurs at *U_r_* = 9 and *U_r_* = 10. Owing to the minimal VIV inherent to the whisker model, its vibration is primarily driven by external periodic forces. Therefore, *f*_dominant-Y_* generally remains locked in the vortex shedding frequency of the upstream bluff body (*St* = 0.141). On the other hand, the lift force is influenced by both the external vortex shedding frequency and the vibration of the whisker itself. This complex interaction between the vibration and the vortex shedding leads to a temporary decoupling of *f*_dominant-Y_* and *f*_dominant-FL_*. As shown in the lift amplitude spectrum in Figure 11, after *F_L,rms_* reaches a local peak at *U_r_* = 7, the amplitude of the lift force in the frequency range of *f** ≈ 1.0–1.5 significantly decreases.

Figure 12a presents the mean drag force *F_D,mean_* of the whisker model. Compared to uniform flow, the wake flow generally reduces *F_D,mean_* in most cases, with a more pronounced reduction for larger characteristic dimensions. At higher *U_r_*, *F_D,mean_* in the wake of a 48 mm square cylinder even exhibits a downward trend with increasing *U_r_*. From an overall perspective, as the characteristic dimension of the bluff body increases, the variation range of *F_D,mean_* with *U_r_* decreases. In terms of shape, the average *F_D,mean_* is quite similar for circular and triangular cylinders, measuring 0.315 N and 0.317 N, respectively, while it is relatively lower for square cylinders at 0.287 N. This discrepancy may be attributed to the more significant impact of the negative pressure zone formed downstream of the square cylinder on the whiskers.

Figure 12b shows the RMS drag force *F_D,rms_* of the whisker model. *F_D,rms_* in uniform flow is relatively small, while the wake flow significantly enhances *F_D,rms_* values. Unlike the *F_L,rms_*, *F_D,rms_* increases more smoothly with *U_r_*, following a quadratic trend. Fitting each curve with a quadratic polynomial yields a coefficient of determination (R^2^) greater than 0.98, indicating that *F_D,rms_* can reflect flow velocity information to some extent. Figure 13 intuitively displays the mean *F_D,rms_* for each case. It is observed that the magnitude of *F_D,rms_* follows a similar trend to the *A**_10_, i.e., square cylinder ≈ triangular cylinder > circular cylinder, and larger characteristic dimensions yield higher values than smaller ones (48 mm > 32 mm > 16 mm). This finding suggests that the excitation effect on the *F_D,rms_* also exhibits a strong correlation with the total circulation generated in the wake of bluff bodies.

### 3.2. Machine Learning-Based Shape Classification

#### 3.2.1. Data Preprocessing

The data preprocessing is conducted using the data from *U_r_* = 4–30 to train the LSTM-based classification model. The detailed information in the data is shown in Table 3. The original data duration is 110 s with a sampling frequency of 1000 Hz. We downsampled the original data by selecting one data point every 10 time steps, reducing the sampling rate to 100 Hz. This process helps to increase sample diversity, enrich feature information, and mitigate overfitting [30]. Each original dataset was thus split into 10 new 100-Hz datasets. These processed data are not directly fed into the model in their raw time-series form. Instead, they undergo comprehensive feature extraction before being input to the neural network for training.

Based on the analysis of the vibration response and fluid forces of the whisker model, 8 features were extracted from the new datasets: *A**_10_, *F_L,rms_*, *F_D,mean_*, *F_D,rms_*, *f*_dominant-Y_*, *f*_dominant-FL_*, *a*_Y_*, and *a*_FL_*. Here, *a*_Y_* and *a*_FL_* represent the non-dimensional amplitudes corresponding to *f*_dominant-Y_* and *f*_dominant-FL_* in the amplitude spectrum, respectively. These features were chosen for their capacity to reflect the distinct responses of the whisker under various flow conditions and its interaction with upstream wake vortices. They encapsulate key facets of the whisker’s hydrodynamic behavior, including vibration characteristics, fluid force metrics, and frequency-related information, thereby offering a holistic depiction of the whisker’s engagement with the surrounding fluid environment.

Before feeding these feature values into the model for training, they were normalized to the range of [0, 1]. The final dimension of the total dataset was 1700 × 8. The target categories for LSTM classification include four types: circular cylinder wake (referred to as C hereafter), square cylinder wake (S), triangular cylinder wake (T), and uniform flow (U). Table 4 provides the concepts and formulas for the metrics used in this study to evaluate the overall classification performance of the model.

To evaluate the performance of the LSTM-based classification model, 80% of the feature rows were randomly selected to form the training set, with the remaining 20% allocated to the test set. The training result is illustrated in Figure 14 and Figure 15. As depicted in Figure 14, during the initial stages of training, the cross-entropy loss for both the training and validation sets exhibited a rapid decline, and the accuracy increased swiftly. This indicates that the model was quickly learning from the training data and generalizing well to the validation set. Subsequently, both the loss and accuracy values entered a phase of minor fluctuations within a narrow range, suggesting that the model had reached a relatively stable state in terms of its learning progress. Notably, after the learning rate was adjusted at the 450th epoch, the loss and accuracy metrics remained consistently stable, further highlighting the robustness of the training process. Figure 15 displays the confusion matrices for both the training and test sets. The model successfully identified all wake categories in both sets with a perfect 0% false identification rate (*FIR*) and missed identification rate (*MIR*). To further validate the robustness and reliability of the model, 10 additional training iterations were conducted. In each of these iterations, the model achieved an accuracy of 100%. This consistent performance across multiple training sessions underscores the exceptional accuracy and effectiveness of the LSTM-based classification model in distinguishing between different wake categories.

#### 3.2.2. Flow Velocity Generalization Performance

The above analyses reveal that the correlation between the wake of upstream bluff bodies and the whisker’s vibrational response varies with *U_r_*, exhibiting distinct response characteristics and underlying mechanisms. Consequently, it is essential to analyze the model’s identification performance across different flow velocities. Prior research predominantly centered on classification accuracy and the generalization capability for data within the scope of trained experimental conditions [18]. In contrast, this study strategically removes features at random *U_r_* from the training set and incorporates them into the test set. By doing so, it systematically evaluates the model’s flow velocity generalization performance through varying the test-to-train ratio (*R_s_*). *R_s_* is defined as(3)$$R_{S}=\frac{N_{test}}{N_{train}}\times 100\%$$
where *N_train_* is the training set’s *U_r_* count and *N_test_* is the test set’s. The generalization ability of the model is evaluated by the accuracy on the test set. The higher the accuracy, the better the model’s adaptability and prediction accuracy for data under different flow velocities.

For each *R_s_*, the model is trained following the aforementioned data partitioning procedure, with the process repeated 20 times to ensure statistical reliability. The average classification accuracy for each *R_s_* is subsequently determined by computing the mean value across all repetitions, as shown in Figure 16. It can be observed that as *R_s_* increases, the training set accuracy remains consistently at 100%, while the test set accuracy exhibits a nearly linear decline. This indicates that as the training data becomes sparser in terms of *U_r_*, the model’s generalization ability is weakened. When *R_s_* is small (i.e., with a larger training set), the model successfully learns features across a wider range of velocities, enabling better generalization to unseen velocities in the test set. As *R_s_* increases, the training set size declines and limits the representation of velocity conditions in the training data, leading to progressively worse generalization performance.

Figure 17 illustrates the variation in the F1 score for the test set across four wake identification types as *R_s_* changes. The F1 score is a commonly used comprehensive evaluation metric for assessing the performance of classification models [31]. The results demonstrate that for all four wake categories, the F1 score exhibits a linear decreasing trend with increasing *R_s_*, which shows consistency with the degradation characteristics of the overall model accuracy. Although the F1 score reflects the model’s generalization performance for specific categories, while accuracy characterizes the overall generalization capability, the consistent trends between these two metrics confirm that insufficient training data coverage (increasing *R_s_*) leads to synchronous deterioration of the model’s generalization performance across different levels. However, the F1 scores for square cylinder wake and uniform flow are slightly higher than those for circular cylinder wake and triangular cylinder wake. This indicates that the generalization ability is better for the square cylinder wake and uniform flow.

To evaluate the classification performance of the LSTM-based model in distinguishing between different wake flows, the confusion rate between two types of wake is denoted by *r_A−B_*, as shown in Figure 18. It can be observed that the model has a relatively high confusion rate in identifying the wakes of circular and triangular cylinders. The underlying causes of this phenomenon will be explored in the subsequent discussion. In uniform flow, the whisker model displays minimal *A**_10_, *F_L,rms_,* and *F_D,rms_* and simultaneously fails to exhibit a clear dominant frequency. These features are more pronounced and distinct compared to those in the wake of bluff bodies, resulting in a very low overall confusion rate. The sole exception is the somewhat higher confusion rate observed between the triangular cylinder wake and uniform flow. The cumulative confusion matrix for each *R_s_*, summed over the 20 iterations, is presented in Figure 19.

#### 3.2.3. Feature Significance

When *R_s_* = 0.889(*N_train_* = 9, *N_test_* = 8), we investigate the impact of individual features on model performance by varying the number of input features. With 8 features in total, there are 255 non-empty subsets to evaluate. For each subset, the model is trained 3 times, and the average training and testing accuracies are recorded. To measure the impact of features on the model’s classification performance, an accuracy difference, denoted as Δ*Acc*, is defined as an evaluation metric. The formula is as follows:(4)$$\Delta Acc=Acc_{in}-Acc_{ex}$$

Here, *Acc_in_* represents the average accuracy when the feature (or feature combination) is included. *Acc_ex_* represents the average accuracy when the feature (or feature combination) is excluded. A higher Δ*Acc* for the training set indicates that the feature significantly enhances the model’s classification performance on the training data. This suggests that incorporating the feature allows the model to fit the training data more effectively. Similarly, a higher Δ*Acc* for the test set indicates that the feature significantly enhances the model’s classification performance on unseen data. This suggests that incorporating the feature allows the model to generalize better to new data in practical applications.

Figure 20 shows the Δ*Acc* of the training and test sets for eight features. It is evident that the Δ*Acc* of *F_D,mean_* significantly outperforms other features. Conventionally, it has been thought that in identifying hydrodynamic trails and upstream wakes, the critical factor is the additional small jerky type motion of seal whiskers in response to a single vortex core [32]. This corresponds to pulsation amplitude features in this study, such as *A**_10_, *F_L,rms_*, and *F_D,rms_*. The current findings indicate that the mean drag force (*F_D,mean_*) is critical for distinguishing between different bluff body types, a factor often overlooked in previous studies. The *F_D,mean_* serves as a dominant feature in the classification process, which explains the relatively high confusion rate between triangular and circular cylinders, as their *F_D,mean_* values are closer in magnitude and trend. Moreover, *f*_dominant-FL_* appears to contribute significantly more to generalization than *f*_dominant-Y_*. This observation may be linked to the lock-in phenomenon of *f*_dominant-FL_*, as discussed earlier. It is speculated that this phenomenon could enable *f*_dominant-FL_* to capture more shape-related information compared to *f*_dominant-Y_*. Lastly, *a*_Y_* and *a*_FL_* show minimal Δ*Acc* values across both training and test sets. This suggests that amplitude spectral information provides only limited support for model training and generalization. As illustrated in Table A1, excluding *a*_Y_* and *a*_FL_* still yields high accuracies: the training set accuracy stays at 100%, and the test set accuracy reaches 94.67%.

Table A1 encompasses the average accuracies and Δ*Acc* values for the training and test sets across all feature combinations. Notably, no other feature combination was identified to enhance model performance as markedly as *F_D,mean_*. This observation reinforces the pivotal role of *F_D,mean_* in the classification model.

## 4. Conclusions

This study investigates the vibration and hydrodynamic responses of a harbor seal whisker model in the wakes of bluff bodies with varying shapes and sizes, culminating in their application for target identification by adopting an improved machine learning model. The main conclusions are summarized as follows.

In the wakes of bluff bodies, the whisker model’s vibration amplitude significantly correlates with the total vortex shedding circulation, leading to higher peak amplitudes for square and triangular cylinders than circular ones. Larger bluff body dimensions result in higher peak vibration amplitudes. Additionally, when the model’s dominant vibration frequency nears its natural frequency, vibration amplitude peaks, occurring at lower *U_r_* for smaller bluff body dimensions or higher *St*;In specific regions, the lift force rms value exhibits a local peak followed by a valley, forming an “N-shaped” pattern, with the inflection point closely corresponding to *U_r_* at which the vibration amplitude reaches its peak. Additionally, the dominant frequency of lift exhibits a more pronounced locking phenomenon compared to that of displacement, leading to a temporary decoupling between them;The mean drag force of circular and triangular cylinders is quite similar, while that of square cylinders is relatively lower. The trend of the RMS drag force is similar to that of vibration amplitude, indicating that the excitation effect of drag force amplitude is also closely related to the total circulation in the wake of bluff bodies;The linear decline in the average accuracy of the test set with the increasing ratio of test-to-train set indicates that the model’s generalization ability weakens as the training data becomes sparser in terms of flow velocity. The mean drag force significantly outperforms other features or combinations in enhancing the accuracy of both the training and test sets, playing a crucial and core role in the classification process.

The machine learning-related phenomena observed above were obtained under well-defined experimental conditions using our current model architecture. While these specific findings require validation across broader parameter ranges, the proposed methodology may contribute to performance assessment methodologies in bio-inspired flow sensing research. Further studies with expanded experimental matrices will be necessary to establish the generalizability of these observations. Additionally, given the limitations of current experimental methods in accurately capturing the complex behaviors of elastic seal whiskers, the present study focused on investigating the dynamic responses of a rigid whisker model within a parameter space relevant to real-world conditions. The insights gained from this study provide a foundation for a deeper understanding of the mechanisms underlying the remarkable hydrodynamic sensing capabilities of harbor seals and offer valuable guidance for the development of biomimetic underwater sensors.

## Figures and Tables

**Figure 1 biomimetics-10-00534-f001:**
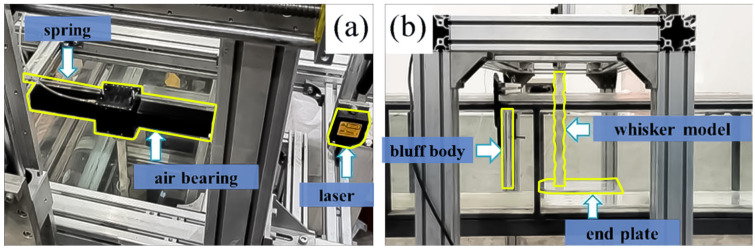
Experimental setup of the test section: (**a**) Top view, (**b**) Side view.

**Figure 2 biomimetics-10-00534-f002:**
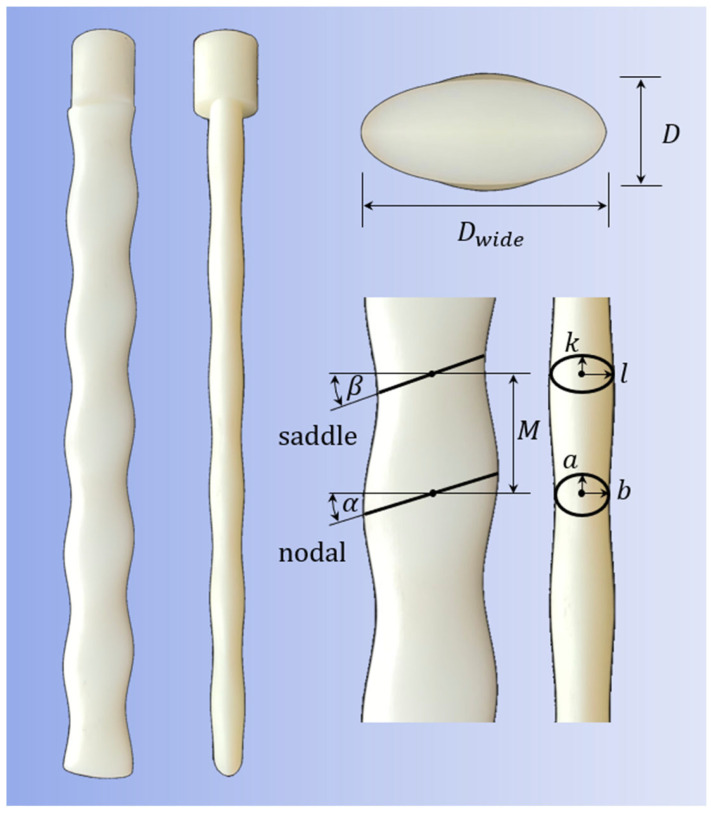
Geometric characteristics of the harbor seal whisker model.

**Figure 3 biomimetics-10-00534-f003:**
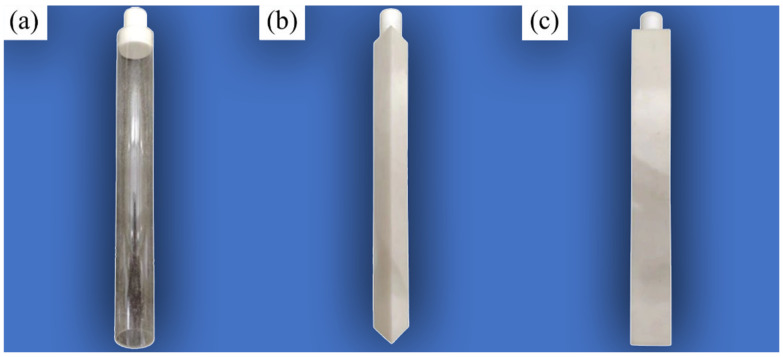
Photographs of the upstream bluff bodies: (**a**) circular cylinder, (**b**) equilateral triangular cylinder, (**c**) square cylinder.

**Figure 4 biomimetics-10-00534-f004:**
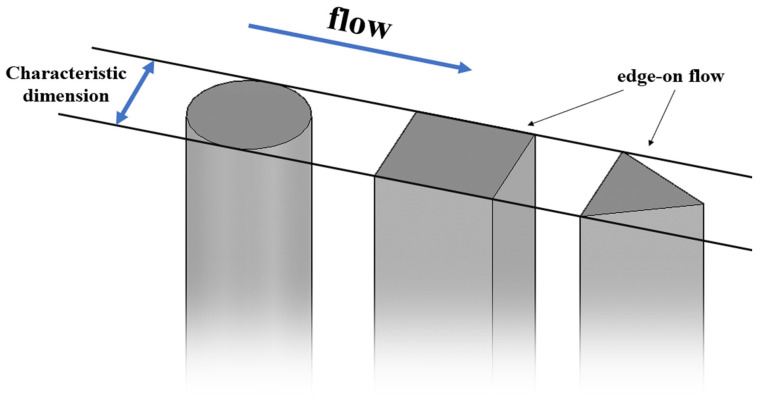
Schematic diagram of the bluff body facing the flow.

**Figure 5 biomimetics-10-00534-f005:**
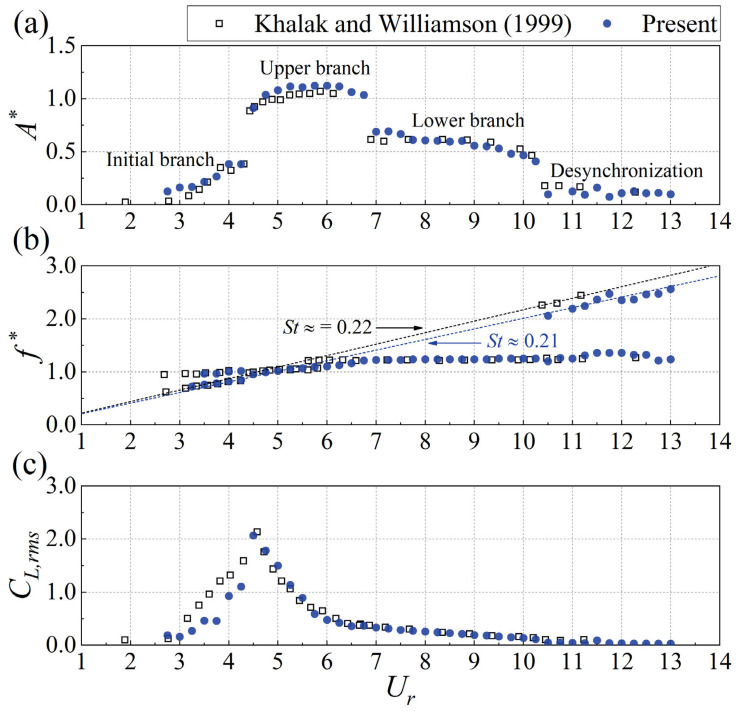
Comparison of (**a**) maximum amplitude, (**b**) vibration frequencies, and (**c**) rms lift coefficients for the vortex-induced vibration of an isolated circular cylinder [20].

**Figure 6 biomimetics-10-00534-f006:**
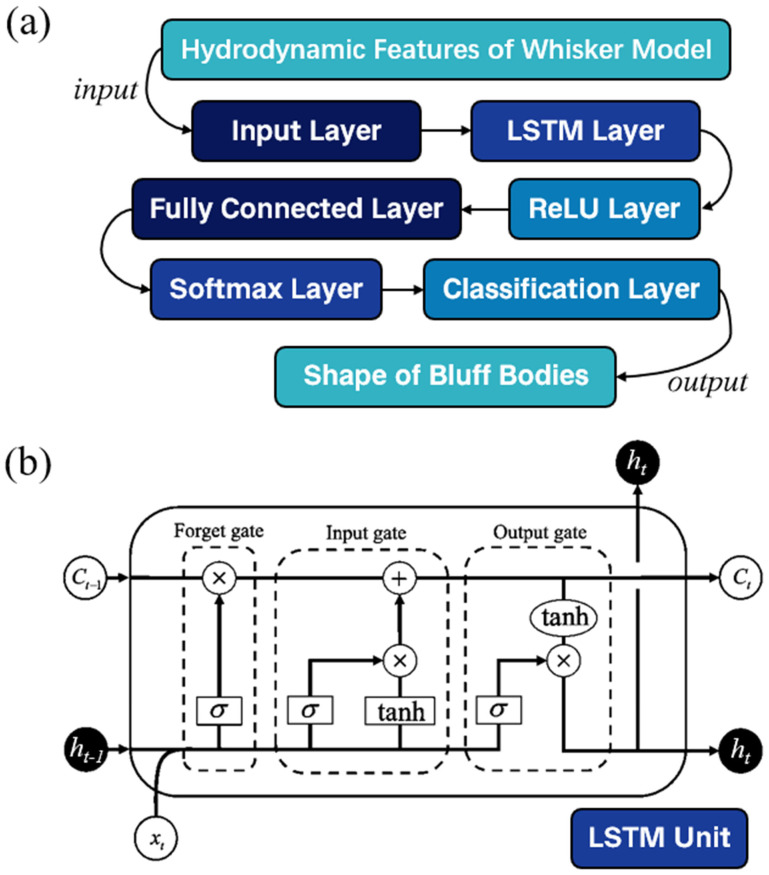
The structure of the (**a**) LSTM-based classification model and (**b**) LSTM unit.

**Figure 7 biomimetics-10-00534-f007:**
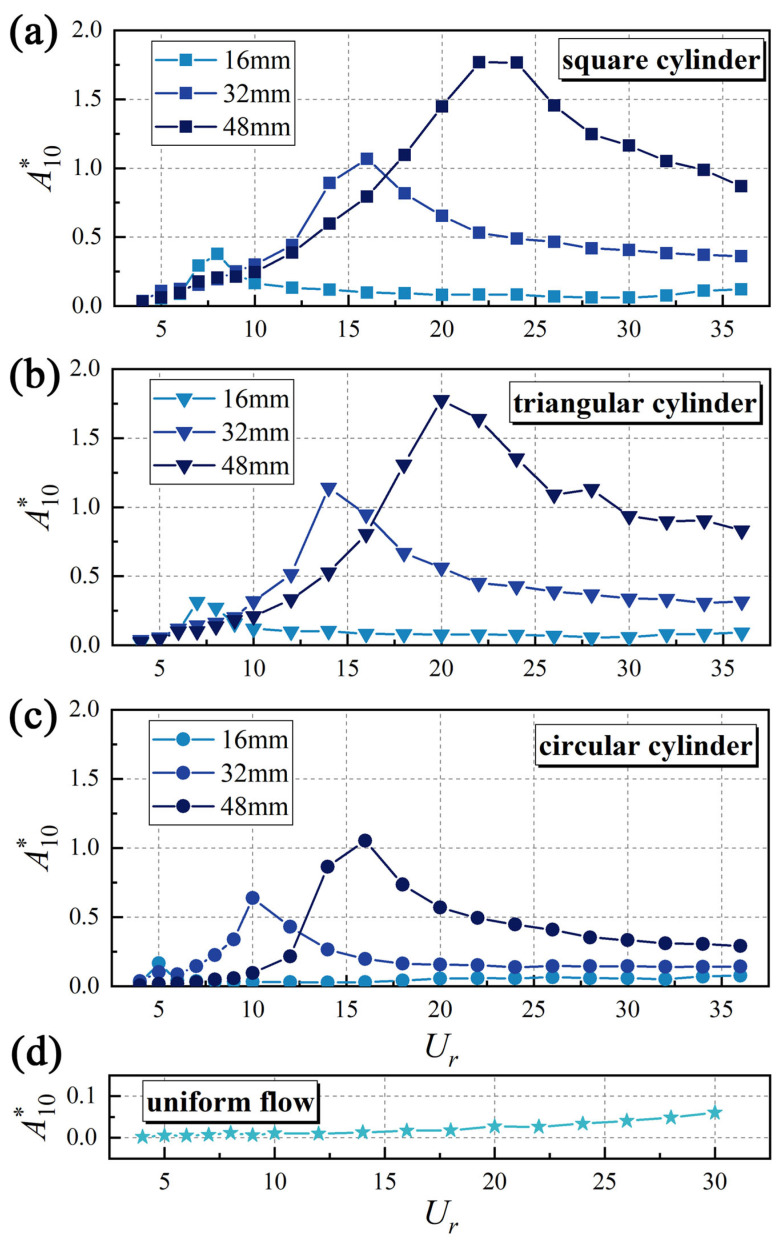
The trend of *A**_10_ with *U_r_* in the wake of the (**a**) square cylinder, (**b**) triangular cylinder, (**c**) circular cylinder, and (**d**) uniform flow.

**Figure 8 biomimetics-10-00534-f008:**
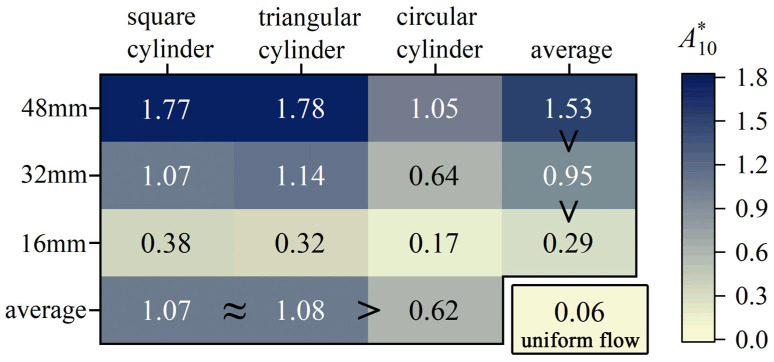
A summary plot of the peak values of *A**_10_.

**Figure 9 biomimetics-10-00534-f009:**
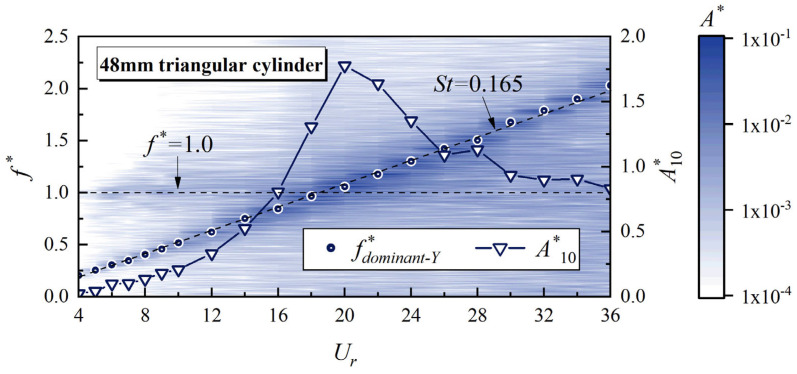
*A**_10_, *f*_dominant-Y_*, and vibration amplitude spectrum of the whisker model in the wake of a 48 mm triangular cylinder.

**Figure 10 biomimetics-10-00534-f010:**
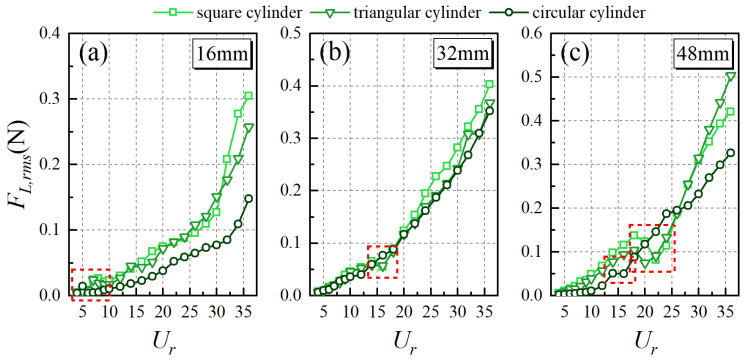
The trend of *F_L,rms_* with *U_r_*. Characteristic dimension: (**a**) 16 mm, (**b**) 32 mm, (**c**) 48 mm.

**Figure 11 biomimetics-10-00534-f011:**
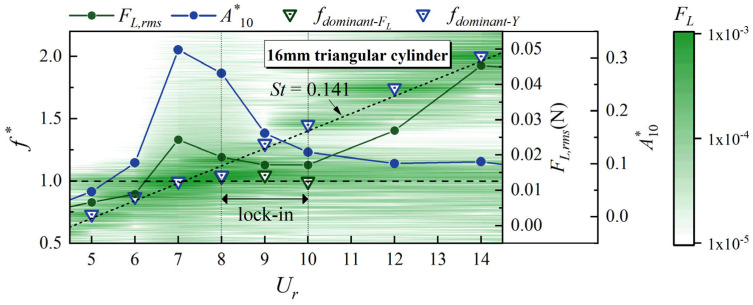
*F_L,rms_*, *A**_10_, *f*_dominant-FL_*, *f*_dominant-Y_*, and lift amplitude spectrum of the whisker model in the wake of a 16 mm triangular cylinder.

**Figure 12 biomimetics-10-00534-f012:**
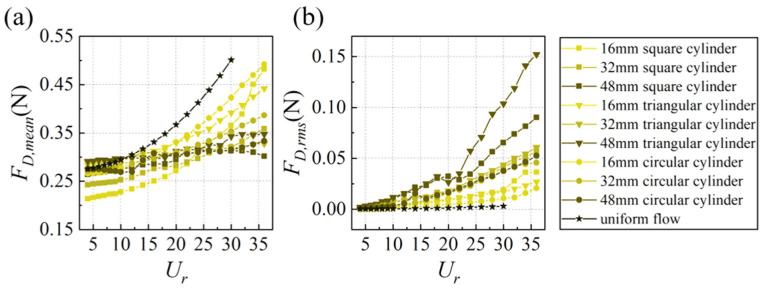
The trend of (**a**) *F_D,mean_,* (**b**) *F_D,rms_* with *U_r_*.

**Figure 13 biomimetics-10-00534-f013:**
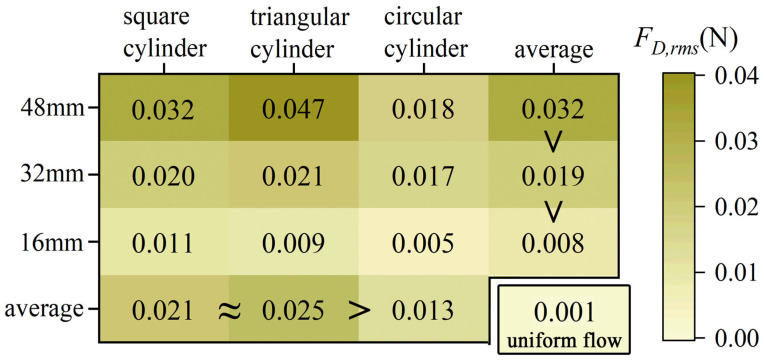
A summary plot of the mean values of *C_D,rms_*.

**Figure 14 biomimetics-10-00534-f014:**
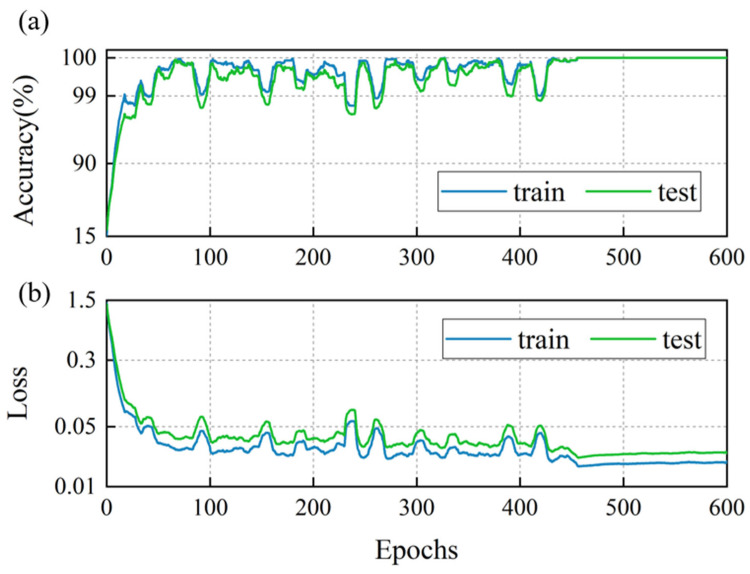
The LSTM-based classification model trained on an 80% random training set: (**a**) accuracy and (**b**) cross-entropy loss.

**Figure 15 biomimetics-10-00534-f015:**
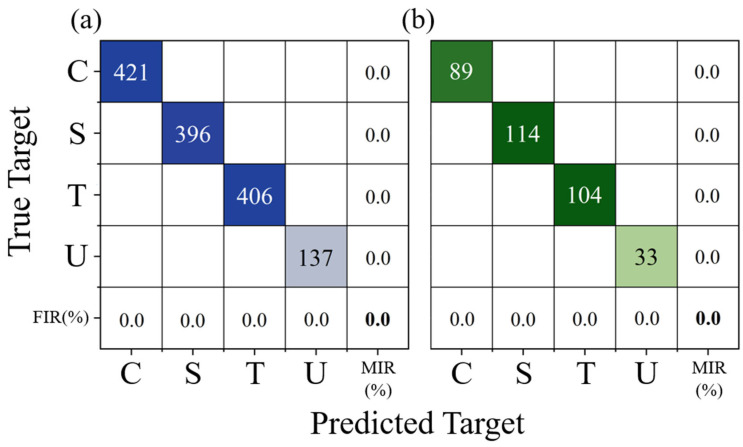
The confusion matrix of the LSTM-based classification model trained on an 80% random training set: (**a**) training and (**b**) testing.

**Figure 16 biomimetics-10-00534-f016:**
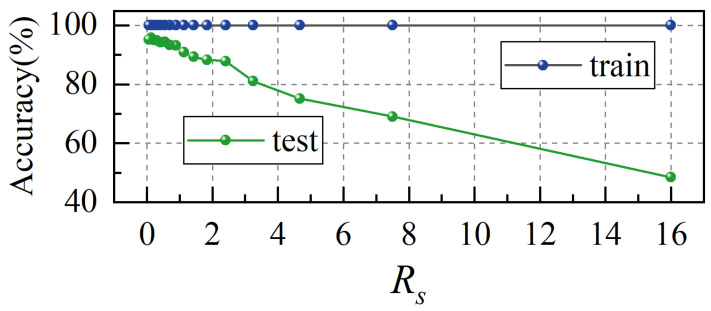
The average accuracy of the training and test sets under different *R_s_*.

**Figure 17 biomimetics-10-00534-f017:**
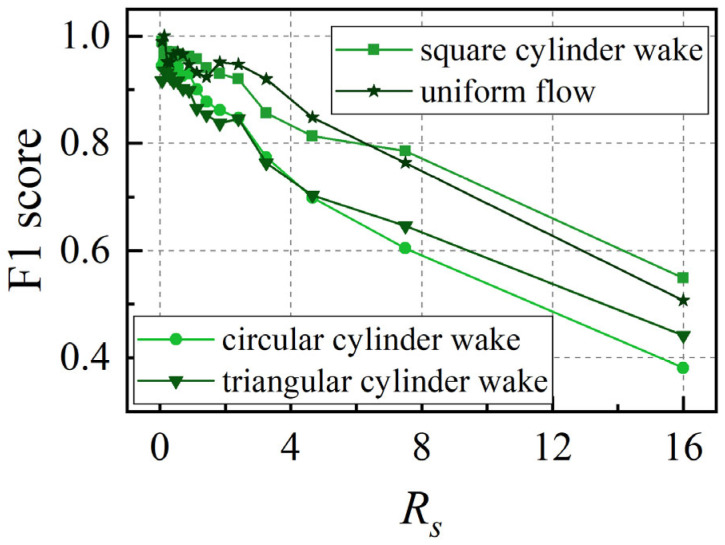
Trend of the F1 score for the test set of four types of wake identification with *R_s_*.

**Figure 18 biomimetics-10-00534-f018:**
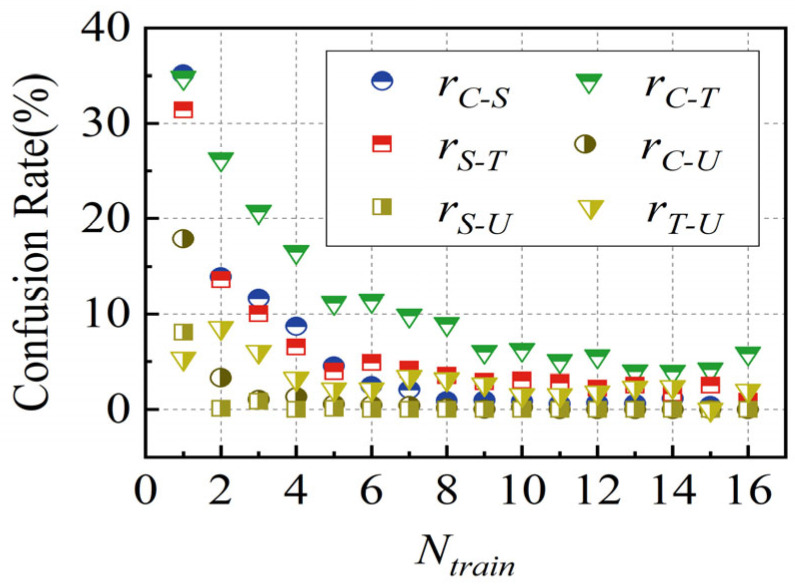
Confusion rate between two types of wake.

**Figure 19 biomimetics-10-00534-f019:**
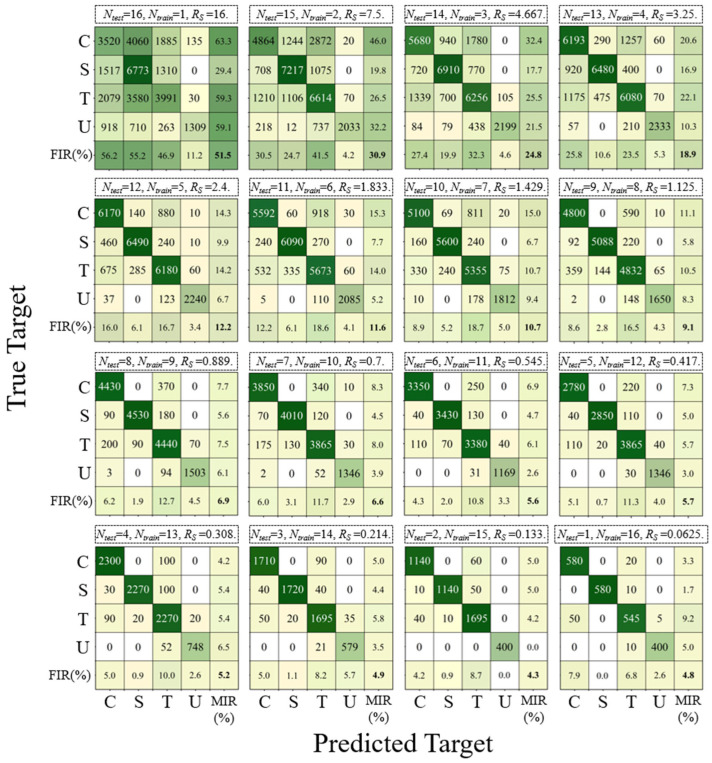
Cumulative confusion matrix for each *R_s_*.

**Figure 20 biomimetics-10-00534-f020:**
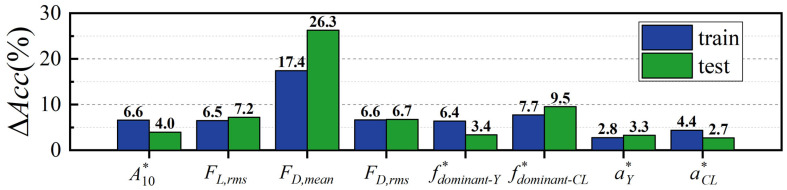
Δ*Acc* of the training and test sets for 8 features.

**Table 1 biomimetics-10-00534-t001:** Parameters in water flume experiments.

Parameter	Value
Natural frequency in still water (*f_n_*)	0.751 Hz
Mass ratio (*m**)	10.29
Inflow velocity (*U_∞_*)	0.048–0.430 m/s
Reduced velocity (*U_r_*)	4.0–36.0
Reynolds number (Re)	760–6840
Structural damping ratio (*ζ*)	0.008

**Table 2 biomimetics-10-00534-t002:** The parameters of the LSTM-based classification model.

Parameter		Value
Hidden units in the LSTM layer		100
Output classes in the fully connected layer		4
Max epochs		600
Learn rate	Schedule	‘piecewise’
	Drop factor	0.1
	Drop period	450
	Initial learning rate	0.075

**Table 3 biomimetics-10-00534-t003:** Detailed information on the data for training the LSTM-based classification model.

Reduced Velocity (*U_r_*)	Wake Categories	
	Bluff Body Shape	Characteristic Dimension
4, 5, 6, 7, 8, 9, 10, 12, 14, 16, 18, 20, 22, 24, 26, 28, 30.	circular cylinder, triangular cylinder, square cylinder.	16 mm, 32 mm, 48 mm.
	uniform flow	
17 in total	10 in total	

**Table 4 biomimetics-10-00534-t004:** The metrics for assessing classification performance of the model.

Metrics	Definition
True Positives (TP)	The number of samples correctly identified as Wake A.
False Negatives (FN)	The number of samples that are actually Wake A but were not correctly identified as such.
False Positives (FP)	The number of samples incorrectly identified as Wake A when they are not Wake A.
True Negatives (TN)	The number of samples correctly identified as not being Wake A.
Accuracy	$\mathrm{Accuracy}=\frac{TP+TN}{TP+TN+FP+FN}\times 100\%$
Confusion Rate	$\mathrm{Confusion}\;\mathrm{Rate}=\frac{FP+FN}{TP+TN+FP+FN}\times 100\%$
Precision (*P*)	$P=\frac{TP}{TP+FP}\times 100\%$
Recall (*R*)	$R=\frac{TP}{TP+FN}\times 100\%$
False Identification Rate (*FIR*)	$FIR=\frac{FP}{TP+FP}\times 100\%$
Missed Identification Rate (*MIR*)	$MIR=\frac{FN}{TP+FN}\times 100\%$
F1 score	$F1\;\mathrm{score}=2\times \frac{P\times R}{P+R}$

## Data Availability

Dataset available on request from the authors.

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
