# Peer review of "Hydrodynamic Responses and Machine Learning-Based Shape Classification of Harbor Seal Whiskers in the Wake of Bluff Bodies"

_biomimetics, 2025, doi:10.3390/biomimetics10080534_

Round 1

Reviewer 1 Report

Comments and Suggestions for Authors

1. In the vibration spectrum presented in this study, the vibration frequency is normalized by the natural frequency of the whisker model. What is the exact value of this natural frequency?
2. What is the mass ratio of the whisker model, and what is the rationale behind selecting this particular value? 
3. Regarding the flow velocity generalization performance discussed in Section 3.2.2, is the test-to-train ratio ($R_s$) manually specified by the authors, or is it automatically determined during the training or testing process? Please elaborate on how this ratio is selected and how it influences the model’s generalization capability. 
4. In Figures 16 and 17, the accuracy and F1 scores are relatively low in some cases—falling below 50%, for example. The authors are encouraged to further explain the performance of the present chosen model and the corresponding reasons.

Reviewer 2 Report

Comments and Suggestions for Authors

Seals, whose diet consists mainly of fish, hunt for it using their whiskers as a kind of sensor. The whiskers detect even the slightest vibrations in the water. With the help of vibrissae, seals can hunt not only during the day, but also at night, and even in murky waters with virtually zero visibility. Scientists believe that by understanding the structure and operating principle of seal whiskers, with which they can sense fish at a fairly large distance and even determine their size, it will be possible to create new high-precision passive analog sensors for ships and submarines.

As of 2017, it was known that no existing detection system used by the military was capable of the sensitivity of seal whiskers. Passive systems detect noises emitted by enemy ships. Passive detection systems are considered the most promising, since they do not emit signals during operation that could be used to detect them. In the non-military sphere, such new types of underwater sensors will find applications in environmental research tasks and will help track the movements of fish and other sea creatures.

Thus, the research topic stated in the work is extremely promising.

The aim of the work is to develop a method for recognizing the shapes of the upstream bluff bodies based on the vibration response of a model seal whisker using a neural network. The work combines both theoretical constructs known from scientific literature and experimental studies conducted by the authors themselves, as well as neurocomputing tools.

The authors demonstrate agreement between their experimental results and those known from the literature, identifying key elements of the system's behavior such as resonance and bifurcation. Below are presented detailed results of an experimental study of average values in the flow and deviations from the average. The experiments are carried out for different flow velocities and different sizes of bodies. The results of the modeling are expressed both in graphical form and in the form of tables. It is worth noting that the corresponding numerical and graphical data are unfussy, clear and very convenient for analysis.

This is followed by a detailed description of the method for recognizing body features using a trained neural network and a comparison of the results of computer and experimental modeling. The resulting error (Fig. 20) ranges from a few percent to approximately 50%, which seems quite acceptable.

Overall, the work leaves an impression of solidity, the presentation is competent and, with some exceptions (see comments below), clear. The Introduction is quite complete and allows you to quickly familiarize yourself with the issues. The Abstract and Conclusion fully reflect the content of the work. The list of references is also quite extensive and corresponds to the essence of the work.

The results of the study appear reasonable and physically sound.

There are some comments on the way the material is presented and interpreted.

1.In formulas (1), (2) the parameters $St, L, T$ are used, which are not defined in the text. Obviously, these are the Strouhal number, the cylinder length and the pulsation period, respectively. For ease of reading, the meaning of these parameters should be indicated. In addition, I did not find mathematical definitions of the dimensionless quantities, mass ratio $m*$, damping ratio $\zeta$ (Lines 157, 158) and frequency $f*$ (Fig.5 and further). The reader feels much more comfortable when the text of the article is autonomous and there is no need to refer to the cited works.

2.The authors claim that the experiment used `an elastically supported, single-degree-of-freedom whisker model’. However, a description of the nature of the material from which the whisker model is made, and therefore its elastic properties, is omitted. In addition, it is not explained how the `single-degree-of-freedom' oscillations of the sample are understood. A more detailed description of the whisker model is needed.

3.The authors argue that 'In the wake flow fields of triangular and circular cylinders, the mean drag is quite similar.' (Lines 19,20). It turns out that vibrations from square cylinders differ significantly from vibrations from triangular and circular cylinders, respectively. This feature is clearly visible, for example, in Fig. 17. It is unclear why this discrepancy occurs, since the square occupies an intermediate position in the process of approximating a circle with polygons between the initial approximation (triangle) and the final one (circle). The authors should explain the nature of this phenomenon.

Conclusion

The article may be accepted for publication after minor revision.
